# Impact of the Design of Walls Made of Compressed Earth Blocks on the Thermal Comfort of Housing in Hot Climate

**Césaire Hema [1,2,\*], Adamah Messan [1], Abdou Lawane [1] and Geoffrey Van Moeseke [2]**

[1] Laboratoire Eco-Matériaux et Habitats Durables (LEMHaD), Institut International d'Ingénierie de l'Eau et de l'Environnement (Institut 2iE), Rue de la Science, Ouagadougou 01 BP 594, Burkina Faso; adamah.messan@2ie-edu.org (A.M.); abdou.lawane@2ie-edu.org (A.L.)

[2] Architecture et Climat, Université Catholique de Louvain (UCLouvain), Place du Levant 1, 1348 Ottignies-Louvain-la-Neuve, Belgium; geoffrey.vanmoeseke@uclouvain.be

\* Correspondence: cesaire.hema@2ie-edu.org or cesaire.hema@gmail.com

**Abstract:** This study investigated the impact of the design of wall systems, mainly made of compressed earth blocks (CEB), on the indoor thermal comfort of naturally ventilated housing in hot climates of Burkina Faso. Conventional housing was modeled and calibrated using the WUFI® Plus dynamic simulation tool based on typical field surveys and the literature. This allowed testing the ability of different wall designs to impact thermal comfort. Thermal discomfort was assessed through an adaptive approach and was based on the annual weighted exceedance hours of overheating. Six designs of walls made of CEB and other locally available materials were simulated and compared to those made of classical hollow concrete blocks. The results of the simulation reveal that the profiles of thermal discomfort vary depending on the wall designs and building spaces. Thus, the wall made, from the outside toward the inside surface, of plywood of 2 cm, an insulation layer of 5 cm and a CEB layer of 29 cm thickness is the most suitable for an annual reduction in overheating for the living room. Regarding the bedroom, the most suitable wall is made of a 14 cm CEB layer, 5 cm insulating layer and 2 cm wood layer from the outside toward the inside surface.

**Keywords:** compressed earth block; building thermal simulation; earthen material; thermal comfort; wall design in hot climate; weighted exceedance hour

## 1. Introduction

The energy performance of buildings is at the heart of policies in developed countries. Guidelines and standards are developed for this purpose to help designers and building practitioners reduce energy consumption at all stages of construction [1]. Indeed, buildings consume 30 to 40% of the primary energy on the global scale for maintaining a comfortable indoor climate [2]. These percentages can be much higher in some southern countries. Throughout a case study in Burkina Faso, Coulibaly [3] showed that nearly 76% of the energy consumption in buildings is used to maintain an acceptable comfort environment. This reveals the inadequacy between the building design and the climate. The directives put in place for energy efficiency and savings are generally based on a reduction in the heat transfer coefficient of the envelope which is suitable for air-conditioned buildings.

In regard to buildings without air conditioning and a mechanical ventilation system, i.e., a naturally ventilated building—which is the case of most dwellings in Burkina—the bioclimatic approach presents the thermal mass as a suitable method to reduce thermal discomfort due to excessive temperatures in a hot climate [4]. Therefore, building materials such as clay bricks, mud bricks, concrete blocks or compressed earth blocks (CEB), which have a high heat capacity, are recommended. These materials

absorb heat from solar radiation at a slower rate and are very effective in countering rapid heat transfer [5].

However, providing the walls with a thermal resistance layer can allow for reducing the energy demand for cooling and therefore has an impact on the thermal comfort of the occupants [6]. Eben Saleh [7] investigated the effect of insulation material on the thermal performance of a building in a hot-dry climate and showed that better performance was achieved by locating the insulation layer on the outer side of the building envelope. Al-Sanea et al. [8] and Meral Ozel [9] reached the same conclusions. However, depending on parameters such as energy consumption, the reverse construction system, i.e., placing the thermal mass layer outside and insulation layer inside, can be more effective. Bojic and Loveday [10] studied the impact of the relative location of the wall layers. Through thermal simulation on a box with a fixed wall thickness and U-value of 0.142 $W\cdot m^{-2}\cdot K^{-1}$, the authors showed that a wall with thermal insulation at the inner side can reduce energy consumption by up to 40% in the case of intermittent heating. The presence of the insulating layer would not only reduce the variation of indoor temperatures but also restrain the heat transfer with respect to the outside climate. Therefore, special attention should be given to walls provided with thermal resistance in order to take advantage of the cool climate at night in naturally ventilated buildings.

The problem, however, lies in the assessment of an acceptable indoor climate and thermal discomfort. Theories on thermal comfort have, for a long time, been based on a heat-balance model, predicted mean vote (PMV) and percentage of people dissatisfied (PPD) index model [11]. The adaptive theory of thermal comfort, defined in EN 15251 [12] and the ASHRAE 55 standard [13], is developed to take into consideration the psychological and physiological adaptation of the occupants as well as the gap found between the expressions of thermal comfort and the PMV model, particularly in non-air-conditioned buildings [14]. Thus, the present study assesses the performance of the indoor climate of the different types of walls based on the adaptive theory of thermal comfort, especially on the weighted exceedance hours (WEH) as recommended by Silva et al. [15].

Significant improvement of thermal comfort can be achieved through the design of the wall [16]. Emmanuel et al. [17] performed thermal modeling of a building under the climate of Ouagadougou (Burkina Faso) and showed that the parameters of thermal inertia such as time shift and wave damping can be improved using a wall made of double layers and provided with a CEB layer and an air gap. Coulibaly et al. [18] defined the design of walls according to the climatic zones of Burkina Faso based on the bioclimatic diagram. The authors prescribed the high thermal inertia of the walls as being associated with natural ventilation in a dry and semi-desert zone. Rincón et al. [19] showed that the high-inertia earthbag technique leads to a reduction in thermal discomfort throughout the year.

In previous studies, the design of walls was based on a static model of comfort, did not take into account the occupancy patterns and did not highlight the advantage of using an alternative sustainable material instead of conventional hollow concrete blocks (HCB) walls.

The present study aims to investigate the thermal performance of different types of walls and their ability to improve the indoor climate of houses. For this purpose, a building was monitored in order to set up a calibrated thermal model. An adaptive index of discomfort and the occupancy profiles of a building's occupants are taken into account to assess the thermal performance of walls. The objective is to identify the wall systems that would improve thermal comfort compared to conventional walls in urban architecture made of HCB.

## 2. Case Study Building

The architecture of the modeled building is based on local and widespread building practices and the architecture found in the urban areas of Burkina Faso. It is a one story house, F2 type, implemented by local real estate developers and shown in Figure 1. The roof of the building consists, from the top towards the interior spaces, of a steel sheet, a ventilated attic and a suspended ceiling made of plywood. Doors and windows are made of metal louvers and are modeled as glazing that is fully shaded in the simulated software. From the outside to the inside, the walls of the building are made of

2 cm of a cement coating, 15 cm of hollow concrete blocks and 2 cm of a cement coating. The main façade of the building is oriented toward the geographic north.

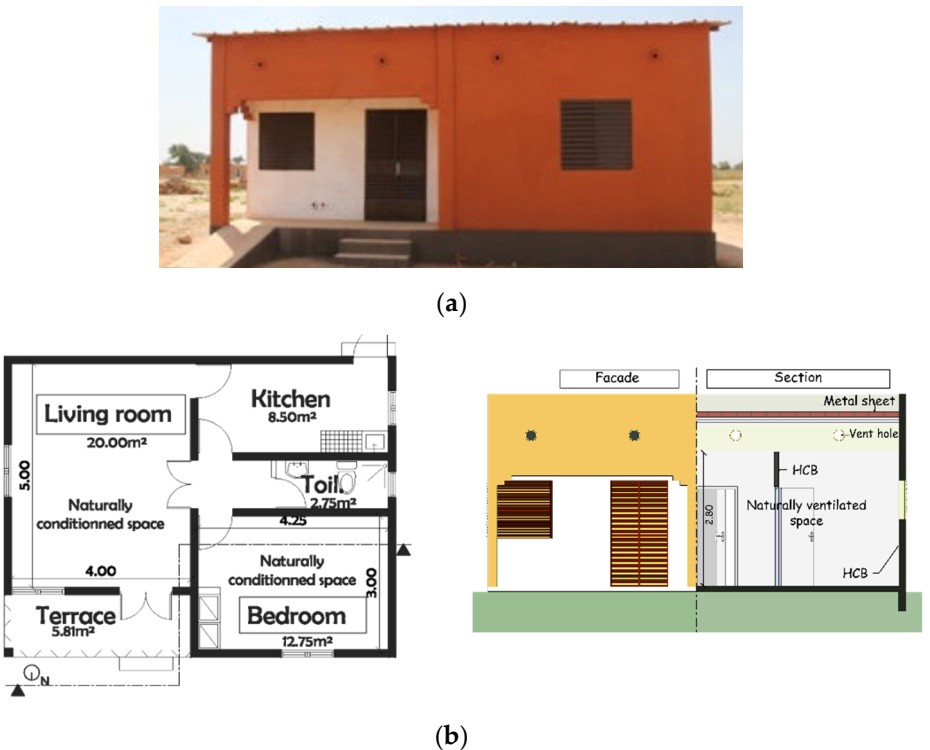

**Figure 1.** Modest and classical urban housing in Ouagadougou: (**a**) picture of the main façade, (**b**) plan and section view.

This type of house, based on hollow concrete blocks, accounts for 65 % of construction in the city of Ouagadougou, the country's capital [20]. Although there are no codes on the thermal performance of buildings in Burkina Faso, vernacular practices emphasize the natural ventilation of inner spaces, the orientation of the building and a covered terrace (see Figure 1a) as principles of indoor climate mitigation. Indeed, the hot climate, shown in Figure 2, requires an adapted architecture because the monthly average temperatures and the maximum temperatures of Ouagadougou vary between 25 and 34 °C and 30 and 40.5 °C, respectively.

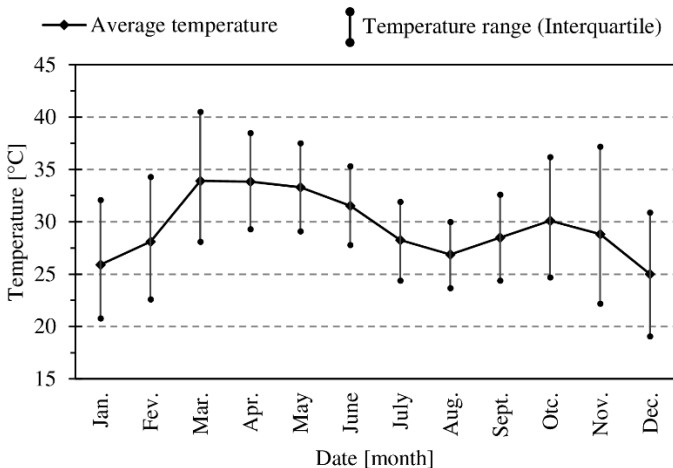

**Figure 2.** Evolution of air temperatures of Ouagadougou based on TMY2 data.

Based on the literature, the thermal properties that have been adopted with regard to the building envelope are presented in Table 1.

**Table 1.** Thermal properties of building materials adopted for the modeling [21].

| System | Material | d [cm] | $\lambda$ [W·m$^{-1}$·K$^{-1}$] | $\rho$ [kg·m$^{-3}$] | c [J·kg$^{-1}$·K$^{-1}$] | $\alpha$; $\varepsilon$ [-] | U-Value [W·m$^{-2}$·K$^{-1}$] |
|---|---|---|---|---|---|---|---|
| Walls | Exterior cement coating | 2.00 | 1.10 | 1700 | 1000 | 0.40; 0.95 | 2.54 |
| | Hollow concrete blocks | 15.00 | 0.80 | 1000 | 1000 | - | |
| | Interior cement coating | 2.00 | 1.10 | 1700 | 1000 | - | |
| Roof | Paint metal sheet (color between red and orange) | 0.15 | 50 | 7800 | 450 | 0.39; 0.95 | 7.14 |
| Doors, windows | Modeled as single glazing completely shaded | - | - | - | - | 0.851 | 5.002 |

## 3. Methodology

### 3.1. Experimental Survey

The experimental measurements of the building were performed from 20 March 2014 to 10 June 2014 by Kabore as a part of his thesis [21]. The building was monitored using temperature sensors type TH from "Waranet Solutions" with accuracies of 0.12 °C. The air temperature was recorded each hour in the living room and in the bedroom. At the time of monitoring, the building was at the end of the construction phase. Therefore, it was not occupied during the data collection and there was no internal heat source. The building was not naturally ventilated (doors and windows were kept closed) during the measurement periods. External measurements were also taken during the monitoring period, including: air temperature; relative humidity; wind speed and direction; and global, direct and diffuse radiation.

In order to avoid the influence of the initial conditions of the model on the predicted results, the period from 20 to 31 March was not included in the calibration. It is believed that this 12-day period was sufficient for the thermal model to converge with the actual measurement parameters. Thus, the calibration used monitoring data from 01 April 2014 to 10 June 2014.

### 3.2. Thermal Model Calibration Criteria

The climatic parameters measured during the monitoring of the building (Figure 1) are temperature and relative humidity. The thermal performance of the model, i.e., its ability to predict the real thermal behavior of the building, is assessed through the normal mean bias error (NMBE), the coefficient of variation of the root mean squared error (CVRMSE) and the correlation coefficient $R^2$ as recommended by ASHRAE [22,23] (Equations (1)–(3)).

$$NMBE = \frac{\sum_1^n \left( x_s^i - x_m^i \right)}{(n-1)\overline{x}_m} \tag{1}$$

$$CVRMSE = \sqrt{\frac{\sum_1^n \left( x_s^i - x_m^i \right)^2}{(n-1)}} \frac{1}{\overline{x}_m} \tag{2}$$

$$R^2 = \frac{\sum_1^n x_m^i x_s^i - n\overline{x}_m\overline{x}_s}{\sqrt{\left(\sum_1^n \left( \left(x_m^i\right)^2 - n\overline{x}_m\right)\left(\sum_1^n \left( \left(x_s^i\right)^2 - n\overline{x}_s\right)\right)}} \tag{3}$$

where $x_m$: measured value; $x_s$: simulated value; $\bar{x}_m$: average measured value; and $\bar{x}_s$: average simulated value.

### 3.3. Setting of the Thermal Model

In order to assess the impact of the wall design on the indoor climate, a modeling and simulation tool, namely WUFI® Plus v2.5.1.0 (from Fraunhofer-Institut fur Bauphysik, Stuttgart, Germany), was used. The WUFI software allowed us to model the heat and moisture transfer and storage in building components. In addition, it simulates the indoor thermal environment and is therefore suitable for addressing issues of thermal comfort and energy consumption in buildings. The mathematical modeling is based on the heat and moisture balance equations described by Künzel [24].

With regard to the calibration process, the thermal model was designed with no inner source of heat (uninhabited spaces) and no ventilation (door and windows closed). Once the thermal model had been calibrated, the initial thermal parameters were extended to take into account the interaction between the occupants and the building. The modeled building is now assumed to be occupied by two adults whose activity type is assumed "Seated, quiet" according to ASHRAE 55 [13], i.e., a metabolic rate of 1 met. The thermal insulation of occupants' clothes is fixed at 0.7 corresponding to an "underwear, shirt, pants, socks, shoes". The house is divided into several living spaces, but this study focused on the living room and the bedroom. The occupancy profiles adopted for both rooms during the weekdays are shown in Figure 3a, while the occupancy profiles during the weekend days are shown in Figure 3b,c.

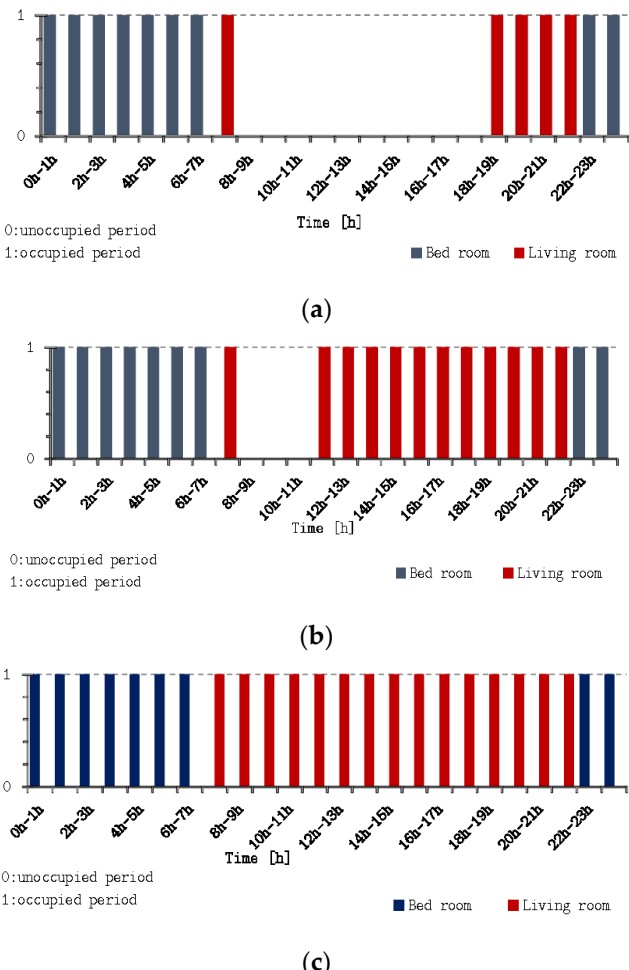

**Figure 3.** Occupancy schedule of the living room and bedroom of the simulated building (**a**) during weekdays, (**b**) on Saturdays and (**c**) on Sundays.

In order to assess the indoor thermal environment of the building over an entire year (instead of the 2 months during the calibration process), the weather data used are the typical meteorological year (TMY2) for the city of Ouagadougou.

In reality, the rates of ventilation and infiltration are governed by the wind and temperature gradient between the exterior and interior air. For the purpose of the study, it is assumed that the adopted rates of ventilation and infiltration are fixed and remain constant for all designs of walls. The rate of air change is set at 0.2 vol·h$^{-1}$ between 8 a.m. and 5 p.m. and 4 vol·h$^{-1}$ between 5 p.m. and 8 a.m. in the living room and bedroom [19]. It is assumed that during the day (8 a.m. to 5 p.m.), the windows are closed to avoid exterior hot air. During the nighttime (5 p.m. and 8 a.m.), the strategy of cooling down the indoor air is based on nocturnal ventilation and therefore the windows are considered to be completely opened. The rate of air change in the ventilated attic of the roof is set by a default at 1 vol·h$^{-1}$. No mechanical ventilation or cooling system is proposed.

### 3.4. Assessment of Overheating Discomfort

The assessment of thermal discomfort is based on a linear regression of the adaptive theory of thermal comfort of the ASHRAE 55 standard [13], as indicated in Equations (4)–(6). The adaptive comfort model is the most suitable theory for the assessment of the indoor climate given that the simulated building is naturally ventilated [25,26].

$$T_{op,sup} = 0.31\,T_{out} + 21.3 \tag{4}$$

$$T_{op,inf} = 0.31\,T_{out} + 14.3 \tag{5}$$

$$T_{out} = (1 - 0.7)\left[ T_{e(d-1)} + 0.7{\cdot}T_{e(d-2)} + \ldots + 0.7^6{\cdot}T_{e(d-7)} \right] \tag{6}$$

where $T_{e(d-i)}$: the main daily outdoor temperature for the previous day.

The operative temperatures outside the upper limit of 80% of acceptability of the model, i.e., high temperatures out of the comfort zone, are counted for one year. This is thermal discomfort due to overheating. It is crucial to assess thermal discomfort only during the occupancy periods of the building spaces. Therefore, the occupancy profiles are important because they are the main difference between different building spaces, i.e., living room and bedroom, in the current case. Therefore, the time during which a room is unoccupied and the operating temperature outside the limits indicated by the adaptive model are not considered in the evaluation of discomfort. In order to consider the intensity of the discomfort, i.e., how high the operative temperature is from the upper limit of comfort acceptability, an index named degree-hour or weighted exceedance hours (WEH) of overheating is defined [13]. The number of hours during which the operative temperature is out of the comfort zone (Equations (4)–(6)), namely ti, is counted and weighted by a factor proportional to the gap between the acceptable limit and the actual operative temperature, Top, sup-Top. The WEH index of overheating is calculated using Equations (7) and (8). Although episodes of discomfort associated with low temperatures exist, they are minimal and less problematic in the hot and dry weather of Ouagadougou and are not considered in this study. The assessment of WEH is directly related to the occupancy profiles of the building spaces (Figure 3). Indeed, the periods during which a room is unoccupied were not considered in the assessment of WEH.

$$WEH = \sum wf = \sum \left( T_{op,sup} - T_{op} \right){\cdot}t_i \tag{7}$$

$$T_{op} = 0.5{\cdot}T_{air} + 0.5{\cdot}T_{int} \tag{8}$$

### 3.5. Walls Design

In regard to the context of Burkina Faso, walls are generally massive and made of adobes, hollow concrete blocks (HCB), lateritic building stones (LBS) or CEB [27]. Firstly, the wall made of HCB and a

cement coating is considered as a reference case because it represents 42.6% and 65% of buildings' walls in urban zones and in the city of Ouagadougou, respectively [20,28]. Then, six other wall types based on CEB are considered in order to assess their ability to impact on thermal comfort. The insulating material combined with CEB is expanded polystyrene. Figure 4 illustrates the different walls considered in the simulation, while Table 2 provides information on dynamic thermal characteristics. The dynamic thermal characteristics of walls were calculated [29] but could have been measured through the procedure given in reference [30].

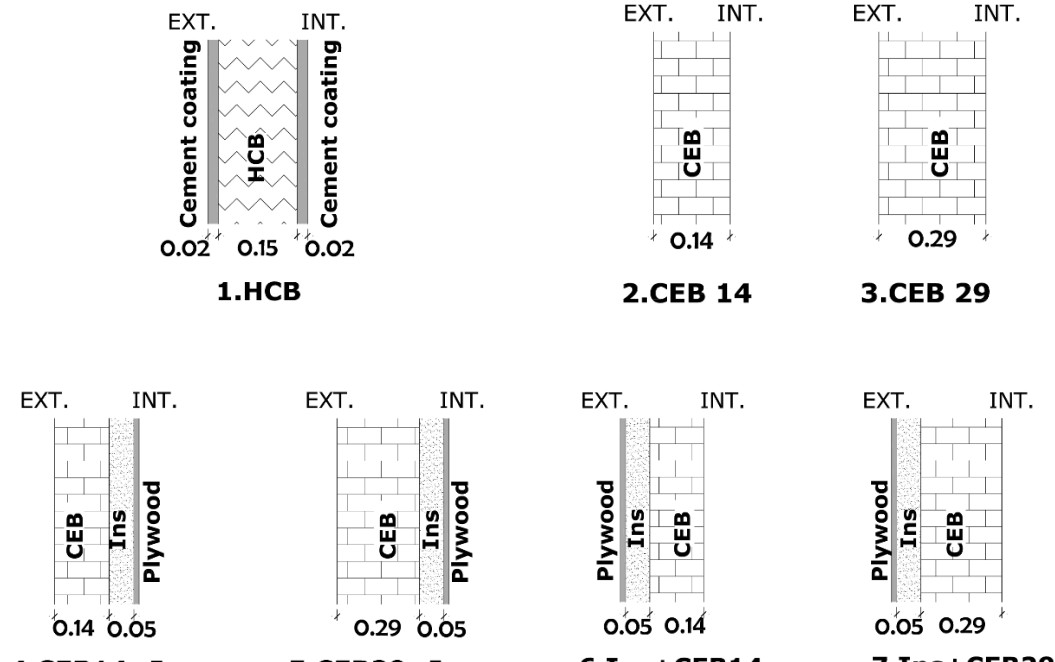

**Figure 4.** Constructive systems of the walls.

**Table 2.** Dynamic thermal characteristics of walls.

|  | Periodic Thermal Transmittance [W·m$^{-2}$·K$^{-1}$] | Time Shift [h] | Decrement Factor [-] |
|---|---|---|---|
| HCB | 1.88 | 4.25 | 0.74 |
| CEB14 | 2.29 | 4.16 | 0.71 |
| CEB29 | 0.65 | 9.05 | 0.297 |
| CEB14 + Ins | 0.29 | 6.19 | 0.5 |
| CEB29 + Ins | 0.078 | 11.03 | 0.15 |
| Ins + CEB14 | 0.17 | 6.92 | 0.3 |
| Ins + CEB29 | 0.049 | 11.56 | 0.09 |

The mass effect of the walls is assessed through the heat capacity and the insulating effect through the thermal transmittance values (U-value). Figure 5 presents the values of these two parameters in comparison to the reference case: the positive and negative values respectively indicate the values which are greater and lesser than those of the reference case. The mass effect on the thermal comfort is assessed through the wall made of CEB29 (CEB layer with 29 cm of thickness). The effect of the improvement in thermal transmittance is assessed through the wall made of CEB14 + Ins (CEB14 + Insulation) and Ins + CEB14. The coupled mass and thermal transmittance effects are assessed through the wall made of CEB29 + Ins and Ins + CEB29.

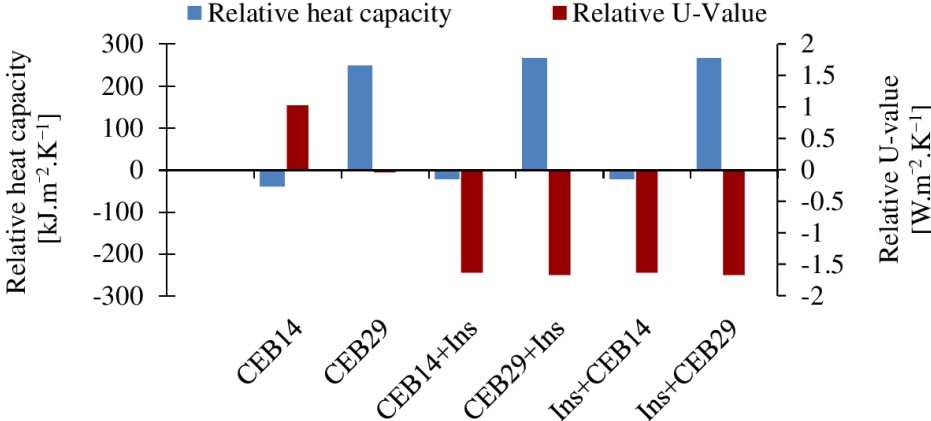

**Figure 5.** Relative heat capacity and thermal transmittance of the walls.

The properties of the classical materials (HCB, cement coating, roof, windows, etc.) were considered from the software database. The review of the literature allowed us to determine the thermal properties of CEB [31–35]. The thermal conductivity of CEB ranges between 0.8 and 1.2 $W \cdot m^{-1} \cdot K^{-1}$, while the density ranges between 1700 and 2200 $kg \cdot m^{-3}$. However, the adopted thermal properties of CEB, i.e., 1.0 $W \cdot m^{-1} \cdot K^{-1}$ and 1920 $kg \cdot m^{-3}$, were based on local blocks [34,35]. The values of the thermal properties of the building components remain the same from one design of a wall to another. Only the position and the thickness of constituent layers vary from one simulation to another.

Figure 6 summarizes the methodology as well as the assumptions adopted.

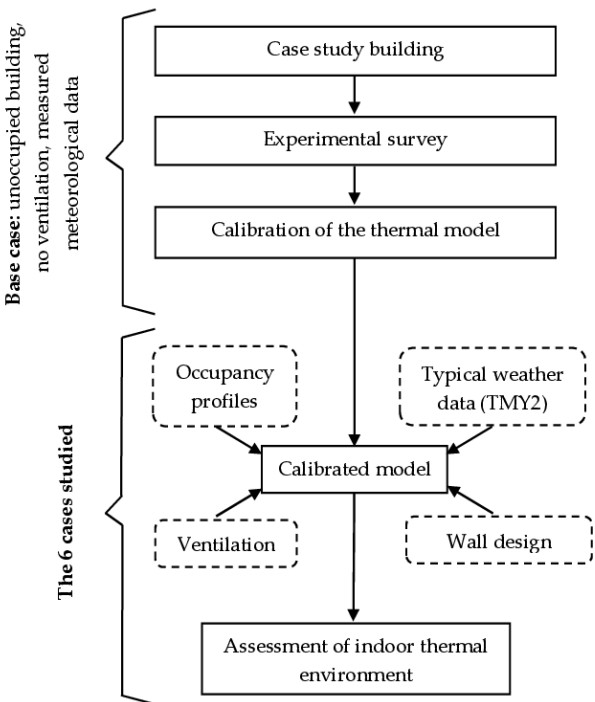

**Figure 6.** Methodological approach of the study.

## 4. Results and Discussion

### *4.1. Calibration of the Model*

Figure 7 shows the measured and predicted temperatures of the living room and bedroom on representative days during the experiment (from 09 to 16 April). The measured temperatures of the living room in Figure 7a and the bedroom in Figure 7b range from 30.9 to 41.5 °C and from 31.5 to 41.1 °C, respectively, while the simulated temperatures range from 30.7 to 41.7 °C and from 31.3 to 41.1 °C. For both rooms, the predicted temperatures are slightly higher than those measured. However, the shape of the two curves is similar and the predicted temperature fit to that measured. Table 3 gives the criteria for validating the thermal model as well as the values obtained for NMBE, CVRMSE and $R^2$. The values of statistical indices fulfil the validation criteria except for the $R^2$ of the living room. The uncertainties of the thermal properties and of the measurements can explain the values of the statistical indices. Nevertheless, since 0.87 is close to 0.90 and the other statistical indices have reached the required values, it is believed that the thermal model can be used to compare the thermal performance of the wall system.

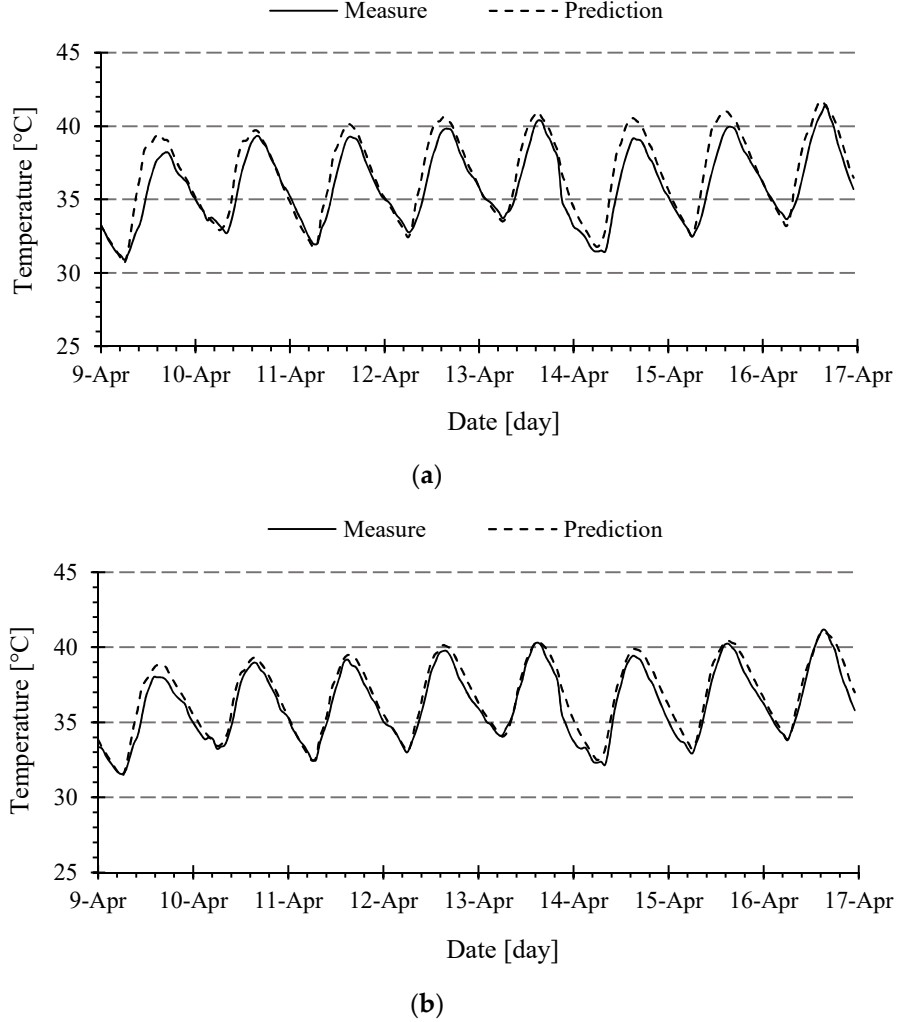

**Figure 7.** Evolution of measured and predicted temperatures in the (**a**) living room and (**b**) bedroom.

**Table 3.** Statistical values of calibration indices.

|  | Required Values for Validation | Temperature of the Living Room | Temperature of the Bedroom |
|---|---|---|---|
| NMBE | <0.101 | 0.036 | 0.025 |
| CVRMSE | <0.30 | 0.048 | 0.035 |
| $R^2$ | ≥0.90 | 0.87 | 0.92 |

*4.2. Impact of Wall Design*

Figure 8 shows the weighted exceedance hours (WEH) over the year as a function of the walls' design compared to the reference case made of HCB: positive and negative values in Figure 8 depend on whether the WEH is higher or lower than that of the reference case, respectively.

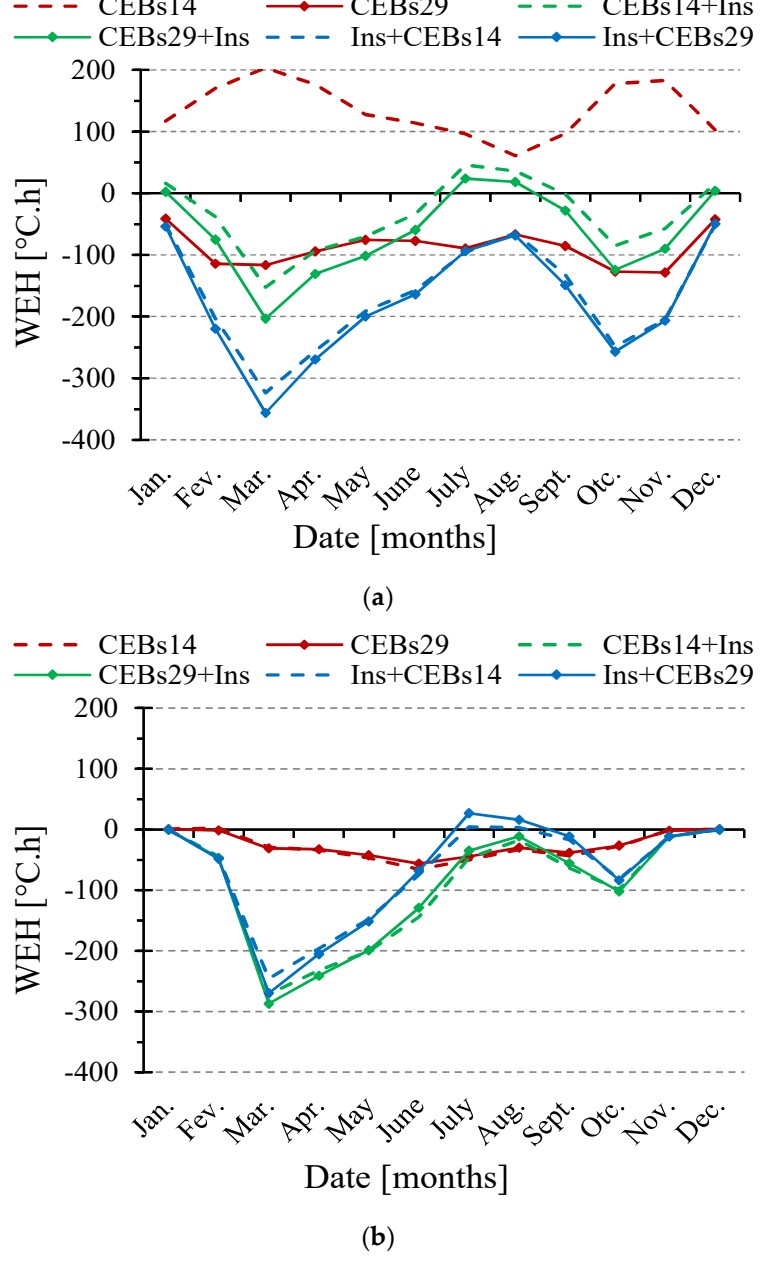

**Figure 8.** Profiles of thermal discomfort of 6 constructive systems compared to the building made of hollow concrete blocks (HCB) for (**a**) the living room and (**b**) the bedroom.

When considering the building space occupied during the day time, i.e., the living room, the increase in wall thermal mass leads to the global reduction in the WEH compared to the building made of HCB walls as illustrated by the graph of CEB29 in Figure 8a. Indeed, the U-value of the wall made of CEB29 and those made of HCB is almost identical (2.17 and 2.21 $W \cdot m^{-2} \cdot K^{-1}$, respectively), while the thermal mass (estimated by multiplying the density, the specific heat and the wall thickness) are 558 and 310 $kJ \cdot m^{-2} \cdot K^{-1}$, respectively. Therefore, the improvement in thermal comfort is only due to the thermal mass effect. When considering the wall made of CEB, having a thickness of 14 cm, the indoor climate is more uncomfortable compared to that of the HCB. The addition of an insulating layer to the CEB layer, i.e., an increase in the thermal resistance of the walls, has a beneficial effect on the level of overheating. However, the best opportunities to reduce the WEH are obtained when the CEB layer is placed at the interior side of the wall. Thus, the wall made, from the outside toward the inside surface, of plywood of 2 cm, an insulation layer of 5 cm and a CEB layer of 29 cm thickness is the most suitable for an annual reduction in overheating for the living room. The results on the room occupied during daytime are similar to those prescribed by the bioclimatic approach [18].

When considering the building space occupied during the night, i.e., the bedroom, all constructive systems performed better than the HCB wall in terms of a reduction in WEH. The least efficient design of a wall is CEB29 and the most efficient are CEB14 + Ins and CEB29 + Ins. Indeed, the only increase in thermal mass (CEB29) is insufficient when the room is occupied during the night but the addition of a thermal resistance layer in contact with the indoor environment allows improving the indoor climate. Thus, the most suitable wall is made of 14 or 29 cm of a CEB layer, 5 cm of an insulating layer and 2 cm of a wood layer from the outside toward the inside surface.

These results show that the constructive systems implemented for the building envelope have an impact on the indoor climate and on the risk of overheating in the hot-dry climate of Ouagadougou. Although the buildings made of HCB have many advantages such as easy implementation, a constructive process well-known by local population, durability and low cost, in terms of indoor climate performance, there are some alternatives based on earth materials. The design of the exterior walls of the building's rooms must consider the occupancy periods and hence should not be unique for the entire envelope of the building. On the one hand, the thermal capacity of the wall, especially when it allows to store the heat from the indoor ambiance, is suitable for the rooms mainly used during the day. Indeed, the thermal mass is an opportunity to store the overheating during the day, especially when it has been efficiently cooled during the night (nocturnal ventilation). This explains why the constructive systems CEB29, Ins + CEB14 and Ins + CEB29 performed better than the HCB walls. On the other hand, when the thermal resistance prevents the heat stored in the thermal capacity layer to reach the indoor environment, this gives opportunities to increase the thermal comfort of the rooms occupied during the night, especially since the cold air brought by the nocturnal ventilation is not used to cool the thermal mass but it is directly available to lower the temperature of the indoor air.

These findings also highlight the complementarity of the thermal resistance and thermal mass of exterior walls in houses without air conditioning systems. Provided with proper ventilation, double-layer walls give more opportunities to improve the indoor climate than the conventional HCB walls.

Research on earthen materials for construction must therefore be directed toward these two properties, i.e., finding materials with high thermal resistance or high thermal mass, instead of materials with both properties. This study also raises the question of the suitable thermal mass layer to be provided in double walls since the improvement of the indoor climate between Ins + CEB14 and Ins + CEB29 or between CEB14 + Ins and CEB29 + Ins is weak.

## 5. Conclusions

The aim of this study was to improve the thermal comfort of houses in a hot-dry climate by using compressed earth blocks instead of hollow concrete blocks in the design of exterior walls. A computer

simulation investigation was carried out in order to assess the weighted exceedance hours of discomfort of six designs of walls, compared to the wall made of hollow concrete blocks. The following conclusions were drawn from this study:

- The thermal comfort of buildings can be improved by the use of alternative sustainable materials such as compressed earth blocks instead of the conventional material used for walls, i.e., hollow concrete blocks;
- Double-layer walls, in terms of thermal properties (i.e., thermal mass and a thermal resistance layer), must be considered in naturally ventilated buildings. They give more opportunity to improve the indoor climate than the hollow concrete blocks walls;
- The appropriate location of the compressed earth blocks layer (i.e., the thermal mass) in double-layer walls depends on the occupancy periods of the rooms. The CEB layer must be placed at the interior side of the wall when the room is mainly occupied during the day and at the exterior side when the room is mainly occupied at night.

For further development, the simulation model should be extended to industrial and administrative buildings. This prospect will allow the evaluation of the performance of the model in terms of energy saving with a defined set point temperature (imposed internal temperature of 28 °C, for example).

**Author Contributions:** Conceptualization: C.H., A.M., A.L., G.V.M.; methodology: C.H., A.M., A.L., G.V.M.; software: C.H., G.V.M.; writing—original draft preparation: C.H., writing—review and editing: C.H., A.M., A.L., G.V.M., funding acquisition: A.M., G.V.M. All authors have read and agreed to the published version of the manuscript.

**Funding:** This research was funded by "Académie de Recherche et de l'Enseignement Supérieur" of the "Fédération Wallonie-Bruxelles (Belgium), Commission de la Coopération au Développement (ARES CCD)" as part of an international research and development project "Amélioration de la qualité de l'habitat en terre crue au Burkina Faso (Improving the quality of earthen housing in Burkina Faso) PRD2016 2021".

**Acknowledgments:** The authors thanks Madi Kabore for providing the raw data of the experimental building. The first author thanks Philbert Nshimiyimana for proofreading the manuscript.

**Conflicts of Interest:** The authors declare no conflict of interest.

## Nomenclature

| | | |
|---|---|---|
| T | temperature | [°C] |
| ti | time step | [h] |
| WEH | weighted exceedance hour | [°C·h] |
| wf | weighted factor | [-] |
| *Subscripts* | | |
| air | related to air | |
| int | related to interior surface of wall | |
| inf | related to the lower limit of acceptability | |
| sup | related to the upper limit of acceptability | |
| op | related to operative temperature | |
| out | related to mean outdoor temperature | |

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
