# Peer review of "Impact of the Design of Walls Made of Compressed Earth Blocks on the Thermal Comfort of Housing in Hot Climate"

_buildings, doi:10.3390/buildings10090157_

Round 1

Reviewer 1 Report

An interesting paper that has wider interest to some readers. The paper could be improved as follows:

  1. Include explicit aims and objectives in Section 1.
  2. Check and correct English in Figure (cement misspelt)
  3. Figure 4 is unclear and should be improved. At present it looks as though there are negative values for U and Heat capacity, rather than changes in these properties. This should be corrected; changing figure title as a start would help, but also axis titles too.
  4. Line 179 "mass' not 'masse'
  5. Values for simulation taken from literature review: how difficult was it? what was the range in properties found in literature? how confident you used the correct values?
  6. Include suggestions for next steps in Conclusions.

Author Response

Response to Reviewer 1 Comments

Point 1: Include explicit aims and objectives in Section 1.

Response 1: The aim and objective are clarified in section 1 (lines 81 to 86)

Point 2: Check and correct English in Figure (cement misspelt)

Response 2: The correction has been made on the word “cement” in figure 3

Point 3: Figure 4 is unclear and should be improved. At present it looks as though there are negative values for U and Heat capacity, rather than changes in these properties. This should be corrected; changing figure title as a start would help, but also axis titles too.

Response 3: Figure 4 has been clarified, using relative parameters of heat capacity and U-value. The title is also adapted to be more relevant

Point 4: Line 179 "mass' not 'masse'

Response 4: The word “mass” in line 179 has been corrected

Point 5: Values for simulation taken from literature review: how difficult was it? what was the range in properties found in literature? how confident you used the correct values?

Response 5: The values of thermal properties from the literature has been specified as well as the adopted properties. Based on studies of locally produced bricks, the thermal properties of CEB were defined (lines 209 to 211).

Point 6: Include suggestions for next steps in Conclusions.

Response 6: A paragraph detailing the next steps has been added to the conclusion (line 298 to 301)

Reviewer 2 Report

This manuscript compare the thermal performance of different types of walls based on an adaptive index of discomfort and taking into account the occupancy profiles of building’s occupant. 

Same improvements are required to define better the methodology used.

After the introduction I would introduce the case study (as a separate paragraph from the methodology) explaining how typical dwellings are built in this country, what materials are used and whether there are any building regulations or building codes. Are renewable sources not used (solar panels on the roof could also be a shading system)? I would also indicate what are the climatic conditions with the typical monthly day: Tavg, DT, Isol, vwind, RH, ... and some annual data such as degree days: CDD and HDD.

Some bibliographical references, such as [17], are not complete.

I would include Table 1 in the case study paragraph and add periodic thermal transmittance YIE, attenuation and time shift data (I would calculate these data later for the different proposed stratigraphies).

In the paragraph on the case study it is reported that, in the measurement campaign ,the building had just been built, was not inhabited and was ventilated only for infiltrations. Afterwards you add the internal gains and change the air changes per hour during the day; how can you compare measurements and calculation? Moreover, if the building has just been built, are the walls and floors no longer damp and therefore have different thermal characteristics?

The calculation methodology and the assumptions adopted should be better explained, perhaps with a flow-chart. Even the weather data used, first it is said that they are measured but then you write that the year type TMY2 is used; at least on the three months of measurements, are the weather data comparable?

In the paragraph "walls design" I would insert a table with thermal transmittance, periodic thermal transmittance, attenuation and time shift (ISO 13786:2017 Thermal performance of building components - Dynamic thermal characteristics - Calculation methods) for the different constructive systems.

It would have been interesting to measure the thermal transmittance characteristics (see for example https://www.mdpi.com/2076-3417/10/8/2858/htm). Perhaps this part could be included for future reference.

In Figure 4 I would put the thermal transmittance and thermal capacity values and not the variations with respect to the HCB wall.

What kind of insulation material was used for the different wall alternatives?

Do the data in Table 2 vary over the different months? Or in days with different weather conditions?

Please, correct the units of measurement in lines 220-222 W/m2/K and kJ/K/m2.

Having made the required corrections, amend with discussion of the results and conclusions.

Author Response

Response to Reviewer 2 Comments

Point 1: After the introduction I would introduce the case study (as a separate paragraph from the methodology) explaining how typical dwellings are built in this country, what materials are used and whether there are any building regulations or building codes. Are renewable sources not used (solar panels on the roof could also be a shading system)? I would also indicate what are the climatic conditions with the typical monthly day: Tavg, DT, Isol, vwind, RH, ... and some annual data such as degree days: CDD and HDD.

Response 1: The case study section has been moved after the introduction as recommended. Further information about the local dwellings, the building regulations and codes and the climate was provided (lines 87-103).

Point 2: Some bibliographical references, such as [17], are not complete.

Response 2: The bibliographic reference [17] has been corrected.

Point 3: I would include Table 1 in the case study paragraph and add periodic thermal transmittance YIE, attenuation and time shift data (I would calculate these data later for the different proposed stratigraphies).

Response 3: Table 1 has been moved to the section case study as recommended. For the reference case, the authors chose to specify the thermal parameters that it is possible to measure in the laboratory (Table 1). Regarding the dynamic parameters of the walls, they will be specified in section 3.5.

Point 4: In the paragraph on the case study it is reported that, in the measurement campaign ,the building had just been built, was not inhabited and was ventilated only for infiltrations. Afterwards you add the internal gains and change the air changes per hour during the day; how can you compare measurements and calculation? Moreover, if the building has just been built, are the walls and floors no longer damp and therefore have different thermal characteristics?

Response 4: A clarification regarding the calibration of the model has been made (lines 139-144). The building is considered unoccupied and not ventilated during the model calibration process as was the case during the measurement campaign. Subsequently, once the model is established, assumptions are made regarding a possible occupancy profiles and air ventilation. The relative humidity of materials has an influence on the thermal properties. However, those adopted allowed the model to be calibrated and are within the range of the literature values of dry materials.

Point 5: The calculation methodology and the assumptions adopted should be better explained, perhaps with a flow-chart. Even the weather data used, first it is said that they are measured but then you write that the year type TMY2 is used; at least on the three months of measurements, are the weather data comparable?

Response 5: A flowchart has been added to clarify the methodology and assumptions adopted (figure 6). This flowchart thus specifies the data used in the calibration phase and those used during the evaluation of the impact of wall designs.

Point 6: In the paragraph "walls design" I would insert a table with thermal transmittance, periodic thermal transmittance, attenuation and time shift (ISO 13786:2017 Thermal performance of building components - Dynamic thermal characteristics - Calculation methods) for the different constructive systems.

Response 6: As recommended, a table with dynamic thermal characteristics of walls has been added (table 2)

Point 7: It would have been interesting to measure the thermal transmittance characteristics (see for example https://www.mdpi.com/2076-3417/10/8/2858/htm). Perhaps this part could be included for future reference.

Response 7: Failure to include the measurement of dynamic thermal characteristics of walls in this study, a reference providing information on the measurement procedure has been added

Point 8: In Figure 4 I would put the thermal transmittance and thermal capacity values and not the variations with respect to the HCB wall.

Response 8: Figure 4 provides information on steady state thermal characteristics of walls. The authors would like to keep the information contained in this figure because it allows a simplified comparison of the thermal performance of the studied walls. However the thermal transmittance and other dynamic thermal characteristics of walls are given in table 2.

Point 9: What kind of insulation material was used for the different wall alternatives?

Response 9: The insulation material used is expanded polystyrene. This clarification was provided in the paper (lines 191-192)

Point 10: Do the data in Table 2 vary over the different months? Or in days with different weather conditions?

Response 10: The data in table 3 (previously table 2) are defined for the entire period of the survey. However, the values of statistical indices could be different if the calibration was performed on monthly basis.

Point 11: Please, correct the units of measurement in lines 220-222 W/m2/K and kJ/K/m2.

Response 11: The units in line 220 -222 have been corrected.

Point 12: Having made the required corrections, amend with discussion of the results and conclusions.

Response 12: The discussion and conclusion have been adapted to the corrections made in the paper.
